# Worsening Glycemia Increases the Odds of Intermittent but Not Persistent *Staphylococcus aureus* Nasal Carriage in Two Cohorts of Mexican American Adults

Heather T. Essigmann,[a] Craig L. Hanis,[b] Stacia M. DeSantis,[c] William B. Perkison,[b] David A. Aguilar,[d] Goo Jun,[b] D. Ashley Robinson,[e] Eric L. Brown[a]

aCenter for Infectious Disease, Division of Epidemiology, Human Genetics, and Environmental Sciences, University of Texas Health Science Center, Houston, Texas, USA
bHuman Genetics Center, Division of Epidemiology, Human Genetics, and Environmental Sciences, University of Texas Health Science Center, Houston, Texas, USA
cDepartment of Biostatistics and Data Science, School of Public Health, The University of Texas Health Science Center, Houston, Texas, USA
dDivision of Cardiovascular Medicine, University of Kentucky, Lexington, Kentucky, USA
eDepartment of Microbiology, University of Mississippi Medical Center, Jackson, Mississippi, USA

**ABSTRACT** Numerous host and environmental factors contribute to persistent and intermittent nasal *Staphylococcus aureus* carriage in humans. The effects of worsening glycemia on the odds of *S. aureus* intermittent and persistent nasal carriage was established in two cohorts from an adult Mexican American population living in Starr County, Texas. The anterior nares were sampled at two time points and the presence of *S. aureus* determined by laboratory culture and *spa*-typing. Persistent carriers were defined by the presence of *S. aureus* of the same *spa*-type at both time points, intermittent carriers were *S. aureus*-positive for 1 of 2 swabs, and noncarriers were negative for *S. aureus* at both time points. Diabetes status was obtained through personal interview and physical examination that included a blood draw for the determination of percent glycated hemoglobin A1c (%HbA1c), fasting plasma glucose, and other blood chemistry values. Using logistic regression and general estimating equations, the odds of persistent and intermittent nasal carriage compared to noncarriers across the glycemic spectrum was determined controlling for covariates. Increasing fasting plasma glucose and %HbA1c in the primary and replication cohort, respectively, were significantly associated with increasing odds of *S. aureus* intermittent, but not persistent nasal carriage. These data suggest that increasing dysglycemia is a risk factor for intermittent *S. aureus* nasal carriage potentially placing those with poorly controlled diabetes at an increased risk of acquiring an *S. aureus* infection.

**IMPORTANCE** Factors affecting nasal *S. aureus* colonization have been studied primarily in the context of persistent carriage. In contrast, few studies have examined factors affecting intermittent nasal carriage with this pathogen. This study demonstrates that the odds of intermittent but not persistent nasal carriage of *S. aureus* significantly increases with worsening measures of dysglycemia. This is important in the context of poorly controlled diabetes since the risk of becoming colonized with one of the primary organisms associated with diabetic foot infections can lead to increased morbidity and mortality.

**KEYWORDS** nasal carriage, *Staphylococcus aureus*, diabetes, dysglycemia, intermittent carriage, persistent carraige, spa typing

Address correspondence to Eric L. Brown, eric.l.brown@uth.tmc.edu.

The authors declare no conflict of interest.

*S*taphylococcus aureus is one of the most common residents of the human nasal microbiome (1) and also an opportunistic pathogen of major concern (2). In both the healthy and those with underlying health conditions, *S. aureus* infections can cause

a range of illnesses, including skin and soft tissue infections, endocarditis, osteomyelitis, meningitis, bacteremia, and pneumonia (2). Approximately 37% of the general population is asymptomatically colonized by *S. aureus* (3), although estimates vary widely depending on the population, with carriers of the bacteria more likely than noncarriers to suffer disease and experience recurrent infections (3–5). In the United States, *S. aureus* colonization of those with diabetes has been estimated at around 32% (6) but ranges widely between 17.5% and 56.6% worldwide (5, 7–14) suggesting that individuals with type 2 diabetes may be at increased risk of *S. aureus* colonization (5, 8, 9, 15).

The possibility of increased prevalence of *S. aureus* carriage in those with diabetes is particularly important since diabetics are more prone to infection and have poorer outcomes after treatment than those without dysglycemia (16–21). *S. aureus* is the most commonly isolated pathogen from diabetic foot infections (22, 23) and the most frequent infectious disease complication for individuals with diabetes, a reality complicated by the fact that many infecting strains are highly virulent and harbor antibiotic resistant genes (5, 21, 23–26). Soft-tissue infections, particularly those caused by antibiotic resistant *S. aureus* (MRSA), result in greater health care utilization, treatment failure, and increased health care costs for affected individuals with diabetes (27, 28).

Although exposure to *S. aureus* is common and different anatomic sites can be colonized (e.g., skin, throat, groin, axilla, nose), nasal carriage is routinely used as a primary site to define carriage phenotypes since nasal decolonization of *S. aureus* is followed by decolonization at other sites, suggesting that the nose is a key 'seeding' niche for this organism (29–31). *S. aureus* nasal carriage in humans is commonly divided into three carrier phenotypes, persistent, intermittent, or noncarriers (3, 4, 29, 32, 33), corresponding to those that repeatedly, occasionally, or never carry culturable levels of *S. aureus* in their nares. In the general population, approximately 80% of carriers are intermittently colonized and 20% persistently so (3). Despite the increased risk of *S. aureus* infection in persistent carriers compared to non and intermittent carriers (typically resulting from autoinoculation with their respective carriage strains), the severity of these infections is generally less compared to disease presentation in intermittent and noncarriers (29, 34, 35).

Since individuals identifying as Hispanic experience both a disproportionate burden of type 2 diabetes (36, 37) and suffer from high rates of *S. aureus* infection, treatment failure, and excessive health care costs, particularly due to MRSA-related infections (28, 38 to 41), there is a need to better characterize the role type 2 diabetes status may have on an individual's *S. aureus* carriage phenotype among Mexican Americans. Situated on the Texas-Mexico border, Starr County, TX, is one of the poorest counties in the United States and has a prevalence of type 2 diabetes that far exceeds national averages (42). In the past 2 decades the prevalence of type 2 diabetes has increased 74% in Starr County (42).

The present study examined two populations in Starr County for whom *S. aureus* carriage phenotypes and diabetes status were established: the primary population consisted of unrelated individuals previously enrolled in various genome-wide association studies (GWAS), and the second was a convenience sample enrolled as an *S. aureus* carriage phenotype GWAS replication cohort subsequently used characterize household *S. aureus* carriage profiles (43–45). The present study demonstrates that increasing blood glucose values (percent glycated hemoglobin A1c [%HbA1c] and fasting plasma glucose), measures of diabetes status, significantly increased the odds of intermittent but not persistent nasal carriage of *S. aureus*, suggesting that poorly controlled diabetes is a risk factor for intermittent colonization that may increase the risk of infections with this organism. In addition, the impact of increasing degrees of dysglycemia as well as other inflammatory conditions, including body mass index and serum cholesterol, on intermittent but not persistent *S. aureus* nasal carriage further highlight differences between the determinants of intermittent and persistent nasal carriage.

## RESULTS

**Description of the study cohorts.** There were 1,080 unrelated participants in the primary study and 879 participants enrolled by household in the replication cohort with *S. aureus* nasal carriage phenotype data (Table 1). Ninety-seven (11.04%) participants in the replication cohort were previous participants of the primary study and were excluded from replication study analyses. Additionally, 15 persistent carriers in the replication cohort were excluded due to incomplete *spa*-typing, resulting in 782 participants across 543 households included in the replication cohort analyses. Eight *S. aureus* carriers defined as persistent carriers by laboratory cultures (i.e., two swabs, collected 2 weeks apart were both presumptively positive for *S. aureus*) in the primary and 10 in the replication study were reclassified as intermittent carriers following *spa*-typing results that showed these carriers were carrying distinct strains of *S. aureus* at each time point. Because the primary study was designed to recruit approximately 50% participants with type 2 diabetes while those enrolled in the replication study consisted of a convenience sample of enrollees by household, participants of the primary study were older and had a higher prevalence of diabetes (49.35% [533/1,080]), with prevalent or incident cases classified using %HbA1c, fasting plasma glucose, and 2-h post-load glucose measures or previous diagnosis. As glucose challenge was not performed on those with a fasting plasma glucose above 160 mg/dL, 164 subjects with type 2 diabetes were not assessed for 2-h post-load glucose; consequently, this variable was not considered in any analyses. Primary cohort subjects had lower total serum cholesterol levels, possibly as a consequence of greater use of lipid lowering medications (32.87% of the primary cohort). Data on the use of lipid lowering medications, detailed information on the type of diabetes medication(s) used, if any, fasting plasma glucose, fasting insulin, and body mass index (BMI) were not available in the replication study; however, proportions of traditional risk factors for *S. aureus* carriage (e.g., recent antibiotic use and frequency, recent skin infection, hospitalization of self or a family member, smoking status, and number of children or total residents in the household) by carriage status were comparable across both study populations. Both cohorts were overwhelmingly female (71.11% and 71.99% in the primary and replication populations, respectively) and antibiotic use in both study cohorts was substantial, with nearly 50% of participants in both cohorts (47.52% and 46.66% in the primary and replication cohorts, respectively) reporting the use of at least one antibiotic in the past 12 months.

Distribution of the three carriage phenotypes were comparable in both populations with 70.93% (766/1,080) and 66.62% (521/782) of participants classified as noncarriers, 13.06% (141/1,080) and 18.54% (145/782) as intermittent carriers, and 16.02% (173/1,080) and 14.83% (116/782) as persistent carriers of *S. aureus* in the primary and replication study, respectively (Table 1). Continuous measures of glycemia, both %HbA1c ($P = 0.026$ in the primary cohort; $P = 0.002$ in the replication cohort) and fasting plasma glucose ($P = 0.023$ in the primary cohort; not available in the replication cohort), were associated with *S. aureus* carriage status and so too was any use of diabetes medication in the replication cohort ($P = 0.05$). In the primary study, use of the most common diabetes medications, metformin and sulfonylureas, was not associated with carriage phenotype while the use of dipeptidyl peptidase 4 (DPP4) inhibitors was, with a higher proportion of intermittent carriers using this drug compared to persistent and noncarriers. A less granular perspective was available for diabetes medications in replication cohort, but the use of any diabetes medication was borderline significant ($P = 0.05$) with a higher proportion of intermittent carriers reporting the use of any diabetes medications. In addition to measures of glycemia, fasting insulin, another measure of glucose metabolism, was also significantly different across carriage phenotypes ($P = 0.036$) with the highest mean value occurring in intermittent carriers. Other features that differed by carriage status in the primary study included BMI ($P = 0.057$) and antibiotic use in the past 30 days ($P = 0.027$), with increasing BMI and use of antibiotics both more prevalent in intermittent carriers. BMI was not available in the replication

**TABLE 1** Participant characteristics by *S. aureus* carriage status in the primary (*n* = 1,080) and replication (*n* = 782) cohorts

| Characteristic | Primary cohort | | | | Replication cohort | | | |
|---|---|---|---|---|---|---|---|---|
| | Noncarriers (*n* = 766) | Intermittent carriers (*n* = 141) | Persistent carriers (*n* = 173) | *P* value[b] | Noncarriers (*n* = 521) | Intermittent carriers (*n* = 145) | Persistent carriers (*n* = 116) | *P* value[b] |
| | Mean (SD) or n (%)[a] | | | | Mean (SD) or n (%)[a] | | | |
| Age | 53.18 (13.20) | 53.24 (13.55) | 52.18 (13.52) | 0.657 | 44.28 (14.76) | 40.99 (14.92) | 43.59 (15.10) | 0.062 |
| Gender | | | | 0.247 | | | | <0.001 |
| Male | 215 (28.07) | 38 (26.95) | 59 (34.10) | | 130 (24.95) | 38 (26.21) | 51 (43.97) | |
| Female | 551 (71.93) | 103 (73.05) | 114 (65.90) | | 391 (75.05) | 107 (73.79) | 65 (56.03) | |
| Hemoglobin A1c (%) | 6.47 (1.66) | 6.88 (2.05) | 6.67 (1.81) | 0.026 | 5.81 (1.22) | 6.26 (1.81) | 5.94 (1.54) | 0.002 |
| Normoglycemia (<5.7%) | 327 (42.69) | 51 (36.17) | 66 (38.15) | 0.29 | 344 (66.03) | 89 (61.38) | 77 (66.38) | 0.208 |
| Prediabetes (5.7 ≤ % <6.5) | 181 (23.63) | 30 (21.28) | 44 (25.43) | | 101 (19.39) | 23 (15.86) | 21 (18.10) | |
| Diabetes (≥6.5%) | 258 (33.68) | 60 (42.55) | 63 (36.42) | | 76 (14.59) | 33 (22.76) | 18 (15.52) | |
| Fasting glucose (mg/dL) | 129.02 (54.08) | 143.31 (70.00) | 130.06 (56.75) | 0.023 | NA[c] | NA | NA | |
| Normoglycemia (<100 mg/dL) | 277 (36.35) | 50 (35.71) | 65 (37.79) | 0.137 | NA | NA | NA | |
| Prediabetes (100 ≤ mg/dL < 126) | 236 (30.97) | 31 (22.14) | 54 (31.40) | | NA | NA | NA | |
| Diabetes (≥126 mg/dL) | 249 (32.68) | 59 (42.14) | 53 (30.81) | | NA | NA | NA | |
| Fasting insulin (mIU/L) | 18.73 (21.84) | 24.75 (44.41) | 18.94 (19.02) | 0.036 | NA | NA | NA | |
| Use of any diabetes medication | | | | 0.205 | | | | 0.05 |
| No | 484 (63.43) | 79 (56.03) | 112 (64.74) | | 439 (84.26) | 112 (77.24) | 102 (87.93) | |
| Yes | 279 (36.57) | 62 (43.97) | 61 (35.26) | | 82 (15.74) | 33 (22.76) | 14 (12.07) | |
| Use of metformin | | | | 0.757 | | | | |
| No | 581 (75.85) | 103 (73.05) | 129 (74.57) | | NA | NA | NA | |
| Yes | 185 (24.15) | 38 (26.95) | 44 (25.43) | | NA | NA | NA | |
| Use of sulfonylureas | | | | 0.328 | | | | |
| No | 637 (83.16) | 110 (78.01) | 141 (81.50) | | NA | NA | NA | |
| Yes | 129 (16.84) | 31 (21.99) | 32 (18.50) | | NA | NA | NA | |
| Use of thiazolidinediones | | | | 0.163 | | | | |
| No | 729 (95.17) | 134 (95.04) | 170 (98.27) | | NA | NA | NA | |
| Yes | 37 (4.83) | 7 (4.96) | 3 (1.73) | | NA | NA | NA | |
| Use of insulin[d] | | | | 0.068 | | | | 0.153 |
| No | 743 (97.00) | 133 (94.33) | 171 (98.84) | | 439 (85.74) | 112 (80.00) | 102 (87.93) | |
| Yes | 23 (3.00) | 8 (5.67) | 2 (1.16) | | 73 (14.26) | 28 (20.00) | 14 (12.07) | |
| Use of DPP4 inhibitors | | | | 0.022 | | | | |
| No | 704 (91.91) | 121 (85.82) | 163 (94.22) | | NA | NA | NA | |
| Yes | 62 (8.09) | 20 (14.18) | 10 (5.78) | | NA | NA | NA | |
| BMI (kg/m²) | 32.09 (7.03) | 33.53 (7.30) | 32.82 (6.66) | 0.057 | NA | NA | NA | |
| Total cholesterol (mg/dL) | 173.21 (39.69) | 179.65 (41.13) | 177.17 (37.71) | 0.139 | 204.02 (46.08) | 210.65 (63.74) | 215.34 (59.99) | 0.069 |

**TABLE 1** (Continued)

| Characteristic | Primary cohort | | | | Replication cohort | | | |
|---|---|---|---|---|---|---|---|---|
| | Noncarriers (n = 766) | Intermittent carriers (n = 141) | Persistent carriers (n = 173) | P value[b] | Noncarriers (n = 521) | Intermittent carriers (n = 145) | Persistent carriers (n = 116) | P value[b] |
| | Mean (SD) or n (%)[a] | | | | Mean (SD) or n (%)[a] | | | |
| Use of lipid lowering medications | | | | 0.58 | | | | NA |
| No | 513 (67.23) | 90 (63.83) | 120 (69.36) | | NA | NA | NA | |
| Yes | 250 (32.77) | 51 (36.17) | 53 (30.64) | | NA | NA | NA | |
| Antibiotic use in the past 30 days | | | | 0.027 | | | | 0.408 |
| No use | 656 (86.20) | 109 (77.86) | 151 (87.28) | | 431 (82.88) | 120 (83.33) | 102 (87.93) | |
| Use | 105 (13.80) | 31 (22.14) | 22 (12.72) | | 89 (17.12) | 24 (16.67) | 14 (12.07) | |
| Rounds of antibiotics used in the past 12 mo | | | | 0.302 | | | | 0.042 |
| 0 rounds | 394 (51.71) | 72 (51.43) | 102 (58.96) | | 261 (50.19) | 82 (56.94) | 73 (62.93) | |
| 1 round | 176 (23.10) | 38 (27.14) | 41 (23.70) | | 107 (20.58) | 27 (18.75) | 26 (22.41) | |
| 2 rounds | 89 (11.68) | 15 (10.71) | 17 (9.83) | | 78 (15.00) | 18 (12.50) | 12 (10.34) | |
| 3 or more rounds | 103 (13.52) | 15 (10.71) | 13 (7.51) | | 74 (14.23) | 17 (11.81) | 5 (4.31) | |
| Smoking status | | | | 0.603 | | | | 0.016 |
| Current smoker | 113 (15.07) | 22 (15.83) | 19 (11.24) | | 73 (14.07) | 35 (24.14) | 22 (19.13) | |
| Former smoker | 132 (17.60) | 23 (16.55) | 26 (15.38) | | 95 (18.30) | 25 (17.24) | 28 (24.35) | |
| Never smoker | 505 (67.33) | 94 (67.63) | 124 (73.37) | | 351 (67.63) | 85 (58.62) | 65 (56.52) | |
| Birthplace | | | | 0.843 | | | | 0.463 |
| USA | 300 (39.16) | 58 (41.13) | 71 (41.04) | | 143 (27.45) | 47 (32.41) | 31 (26.72) | |
| Mexico/Other[e] | 466 (60.84) | 83 (58.87) | 102 (58.96) | | 378 (72.55) | 98 (67.59) | 85 (73.28) | |
| Skin infection in the past 6 mo | | | | 0.807 | | | | 0.966 |
| No | 724 (94.89) | 133 (94.33) | 163 (95.88) | | 501 (96.16) | 139 (95.86) | 111 (95.69) | |
| Yes | 39 (5.11) | 8 (5.67) | 7 (4.12) | | 20 (3.84) | 6 (4.14) | 5 (4.31) | |
| Hospitalized in the past 12 mo | | | | 0.702 | | | | 0.599 |
| No | 668 (88.36) | 120 (86.96) | 153 (90.00) | | 486 (93.46) | 132 (91.03) | 108 (93.10) | |
| Yes | 88 (11.64) | 18 (13.04) | 17 (10.00) | | 34 (6.94) | 13 (8.97) | 8 (6.90) | |
| Household member hospitalized in the past 12 mo | | | | 0.578 | | | | 0.569 |
| No | 691 (90.33) | 124 (87.94) | 153 (88.44) | | 469 (90.02) | 128 (88.28) | 107 (92.24) | |
| Yes | 74 (9.67) | 17 (12.06) | 20 (11.56) | | 52 (9.98) | 17 (11.72) | 9 (7.76) | |
| No. in the household | 3.40 (1.69) | 3.32 (1.57) | 3.54 (1.59) | 0.476 | 4.21 (1.86) | 4.32 (1.85) | 4.16 (1.77) | 0.751 |
| No. of children in the home | 1.18 (1.35) | 1.11 (1.34) | 1.22 (1.31) | 0.741 | 1.82 (1.57) | 1.80 (1.50) | 1.77 (1.5) | 0.951 |

[a] Not all values sum to total sample N due to missingness.
[b] Pearson's chi-squared or Fisher's exact test for categorical variables and one-way ANOVA for continuous variables.
[c] NA, measure not available in the replication cohort.
[d] In the replication cohort: Yes, use of insulin alone or in combination with an oral medication.
[e] 1 born in El Salvador and 2 in Honduras in Replication cohort.

cohort, however; in addition to %HbA1c, gender ($P < 0.001$), rounds of antibiotics used in the past year ($P = 0.042$), and smoking status ($P = 0.016$) were all associated with carriage phenotype, with persistent carriers more likely to be men and less frequent users of antibiotics.

**Results from the primary cohort.** Univariable logistic regression was used to assess the effect of primary study subject characteristics on the odds of intermittent and persistent carriage, respectively, compared to noncarriage (Table 2). The odds of intermittent but not persistent carriage were significantly associated with both measures of glycemia considered, with a one-unit increase in %HbA1c and a 10-unit increase in fasting glucose associated with a respective 13% (odds ratio [OR]: 1.13; 95% confidence interval [CI]: 1.03–1.24) and 4% (OR: 1.04; CI: 1.10–1.07) increase in the odds of intermittent carriage compared to noncarriage. Although use of any diabetes medication (e.g., metformin, sulfonylureas, thiazolidinediones, DPP4 inhibitors, or insulin) was associated with an increase in the odds of intermittent carriage, the only medication with a significant association was the use of DPP4 inhibitors; users of this drug had nearly two times (OR: 1.87; CI: 1.09–3.22) the odds of intermittent carriage compared to noncarriage. Persistent carriers were less likely to use DPP4 inhibitors than noncarriers, but the effect was not statistically significant (OR: 0.70; CI: 0.35–1.39). Other markers of chronic metabolic disease, including increasing fasting insulin (OR: 1.06; CI: 1.01–1.13), BMI (OR: 1.03; CI: 1.003–1.052), and total serum cholesterol (OR: 1.04; CI: 1.00–1.09) were also associated with intermittent, but not persistent carriage in univariable analyses, although the association with cholesterol was only borderline significant ($P = 0.05$). No other traditional *S. aureus* risk factors were associated with either carriage phenotype in univariable analyses, excepting recent and frequent use of antibiotics. Recent antibiotic use, defined as use in the past 30 days, was associated with a statistically significant 78% increase in the odds (OR: 1.78; CI: 1.13–2.78) of intermittent carriage, while increased frequency of antibiotic use in the past year was associated with a decreased risk of persistent carriage for each additional round of antibiotics used. The result reached statistical significance in those using three or more rounds of antibiotics in the past 12 months, with a 51% reduction in odds (OR: 0.49; CI: 0.26–0.90) of persistent carriage in those participants.

All variables with Wald *P*-value less than 0.20 were included in multivariable modeling for both the intermittent and persistent carrier outcomes, with fasting glucose and %HbA1c considered separately due to high collinearity between these two measures of glycemia (spearman's rho = 0.75, $P < 0.001$) (Table 2). When comparing persistent carriers to noncarriers, number of rounds of antibiotic used in the past year, the only variable significantly associated in univariable modeling, was also the only variable that met criteria for inclusion in the final model. Interestingly, fasting glucose, but not %HbA1c was selected for inclusion in the respective final intermittent carriage multivariable models, with %HbA1c excluded after adjustment for the use of DPP4 inhibitors. The magnitude of the effect of fasting glucose on intermittent carriage (adjusted OR [aOR]: 1.03; CI: 1.00–1.06) was largely unchanged from univariable modeling after adjustment for the use of DPP4 inhibitors, BMI, total serum cholesterol, and antibiotic use in the past 30 days. The effects of DPP4 inhibitors (aOR = 1.94; CI: 1.08–3.48), BMI (aOR = 1.03; CI: 1.00–1.06), and antibiotic use in the past 30 days (aOR = 1.86; CI: 1.18–2.94) on intermittent carriage was also largely unaffected by adjustment, while the effect of cholesterol increased and attained significance (aOR: 1.06; CI: 1.01–1.11) in the multivariable model. Interactions between all covariates included in the multivariable intermittent carrier model were examined, but none were present.

While it was determined that all continuous predictors should be modeled linearly in the final model, we chose to model and plot adjusted and unadjusted measures of glycemia as splines to more fully appreciate the relationship between these variables and carriage phenotypes. Figure 1A shows the odds of intermittent carriage compared to noncarriage across the spectrum of fasting plasma glucose with fasting glucose of 126 mg/dL (the clinical cut-point for diabetes) as the referent value, adjusted for use of

**TABLE 2** Odds of intermittent and persistent carriage, respectively, compared to noncarriage in the primary cohort: univariable and multivariable models

| Characteristic | Intermittent carriage | | Persistent carriage | |
|---|---|---|---|---|
| | Univariable models | Multivariable model | Univariable models | Multivariable model |
| | Odds ratio (95% CI) | | | |
| Age[a] | 1.004 (0.87–1.15) | | 0.94 (0.83–1.07) | |
| Gender | | | | |
| Female | REF | | REF | |
| Male | 0.95 (0.63–1.42) | | 1.33 (0.93–1.89) | |
| Hemoglobin A1c (%) | 1.13 (1.03–1.24)[b] | | 1.06 (0.97–1.17) | |
| Normoglycemia (<5.7%) | REF | | REF | |
| Prediabetes (5.7 ≤ % <6.5) | 1.06 (0.65–1.73) | | 1.20 (0.79–1.84) | |
| Diabetes (≥6.5%) | 1.49 (0.99–2.24) | | 1.21 (0.83–1.77) | |
| Fasting glucose (mg/dL)[a] | 1.04 (1.01–1.07)[c] | 1.03 (1.0002–1.06)[b] | 1.003 (0.97–1.03) | |
| Normoglycemia (<100 mg/dL) | REF | | REF | |
| Prediabetes (100 ≤ mg/dL < 126) | 0.73 (0.45–1.18) | | 0.98 (0.65–1.46) | |
| Diabetes (≥126 mg/dL) | 1.31 (0.86–1.99) | | 0.91 (0.61–1.35) | |
| Fasting insulin (mIU/L)[a] | 1.06 (1.01–1.13)[b] | | 1.00 (0.93–1.08) | |
| Use of any diabetes medication | | | | |
| No | REF | | REF | |
| Yes | 1.36 (0.95–1.96) | | 0.94 (0.67–1.33) | |
| Use of metformin | | | | |
| No | REF | | REF | |
| Yes | 1.16 (0.77–1.74) | | 1.07 (0.73–1.57) | |
| Use of sulfonylureas | | | | |
| No | REF | | REF | |
| Yes | 1.39 (0.90–2.16) | | 1.12 (0.73–1.72) | |
| Use of thiazolidinediones | | | | |
| No | REF | | | |
| Yes | 1.03 (0.45–2.36) | | insufficient data | |
| Use of insulin | | | | |
| No | REF | | | |
| Yes | 1.94 (0.85–4.44) | | insufficient data | |
| Use of DPP4 inhibitors | | | | |
| No | REF | | REF | |
| Yes | 1.87 (1.09–3.22)[b] | 1.94 (1.08–3.48)[b] | 0.70 (0.35–1.39) | |
| BMI (kg/m²) | 1.03 (1.003–1.052)[b] | 1.03 (1.005–1.06)[b] | 1.01 (0.99–1.04) | |
| Total cholesterol (mg/dL)[a] | 1.04 (1.00–1.09) | 1.06 (1.01–1.11)[b] | 1.03 (0.98–1.07) | |
| Use of lipid lowering medications | | | | |
| No | REF | | REF | |
| Yes | 1.16 (0.80–1.69) | | 0.91 (0.63–1.29) | |
| Antibiotic use in the past 30 days | | | | |
| No use | REF | | REF | |
| Use | 1.78 (1.13–2.78) | 1.86 (1.18–2.94)[c] | 0.91 (0.56–1.49) | |
| Rounds of antibiotics used in the past 12 mo | | | | |
| 0 rounds | REF | | REF | |
| 1 round | 1.18 (0.77–1.82) | | 0.90 (0.60–1.35) | 0.90 (0.60–1.35) |
| 2 rounds | 0.92 (0.51–1.68) | | 0.74 (0.42–1.30) | 0.74 (0.42–1.30) |
| 3 or more rounds | 0.80 (0.43–1.45) | | 0.49 (0.26 –0.90)[b] | 0.49 (0.26–0.90)[b] |
| Smoking status | | | | |
| Never smoker | REF | | REF | |
| Former smoker | 0.94 (0.57–1.53) | | 0.80 (0.50–1.28) | |
| Current smoker | 1.05 (0.63–1.74) | | 0.68 (0.41–1.16) | |

**TABLE 2** (Continued)

| Characteristic | Intermittent carriage | | Persistent carriage | |
|---|---|---|---|---|
| | Univariable models | Multivariable model | Univariable models | Multivariable model |
| | Odds ratio (95% CI) | | | |
| Birthplace | | | | |
| USA | REF | | REF | |
| Mexico/Other[d] | 0.92 (0.64–1.33) | | 1.08 (0.77–1.51) | |
| Skin infection in the past 6 mo | | | | |
| No | REF | | REF | |
| Yes | 1.12 (0.51–2.44) | | 0.80 (0.35–1.81) | |
| Hospitalized in the past 12 mo | | | | |
| No | REF | | REF | |
| Yes | 1.14 (0.66–1.96) | | 0.84 (0.49–1.46) | |
| Household member hospitalized in the past 12 mo | | | | |
| No | REF | | REF | |
| Yes | 1.28 (0.73–2.24) | | 1.22 (0.72–2.06) | |
| No. in the household | 0.97 (0.87–1.09) | | 1.05 (0.95–1.16) | |
| No. of children in the home | 0.96 (0.84–1.10) | | 1.03 (0.91–1.16) | |

[a]OR interpreted as a 10-unit increase.
[b]$P$ value $< 0.05$.
[c]$P < 0.01$.
[d]2 were born in a country other than the US or Mexico.

DPP4 inhibitors, BMI, total cholesterol, and use of an antibiotic in the past 30 days. As in the model with linear fasting glucose (Table 2), there is a significant increase in the odds of intermittent carriage with worsening glycemia. Interestingly, this model suggests an increase in the odds of intermittent carriage for those with low values of fasting plasma glucose, but this result is not statistically significant. For comparison purposes, Fig. 1B models the same relationship for persistent carriers compared to noncarriers with fasting glucose forced into the model along with the only significant predictor of persistent carriage, rounds of antibiotics used in the past year, and depicts the lack of association between persistent carriage across the spectrum of fasting glucose. Fig. S1A and B shows the respective unadjusted plots, while Fig. S2 shows the unadjusted (Fig. 2A and B) and adjusted (Fig. 2C and D) odds of persistent and intermittent carriage, respectively, across the spectrum of %HbA1c.

**Results from the replication cohort.** As participants were enrolled by household, general estimating equations (family: binomial; link: logit) with an exchangeable correlation matrix were used to assess the effect of participant characteristics on odds of intermittent and persistent carriage, respectively, compared to noncarriers in the replication cohort (Table 3). %HbA1c was the only measure of glycemia collected in this cohort and limited information was available regarding the use of specific diabetes medications (i.e., categories included: no use of diabetes medications, use of oral agents, use of insulin, or use of both insulin and oral agents). BMI and fasting insulin were also not available for analyses in the replication cohort. As in the primary cohort, %HbA1c was significantly associated with intermittent (OR = 1.21; CI: 1.07–1.36) but not persistent carriage (OR = 1.08; CI: 0.94–1.25). Use of any diabetes medication (i.e., use versus no use) increased the odds of intermittent carriage (OR = 1.49; CI: 0.95–2.34) and decreased the odds of persistent carriage (OR = 0.73; CI: 0.40–1.34) but neither effect was significant. Similar results were observed when comparing use of insulin alone or in combination with oral diabetes medications or use of oral diabetes agents alone or in combination with insulin. Other variables significantly associated with intermittent carriage in this population included participant age and smoking status; a 10-unit increase in age was associated with a 15% reduction in the odds of intermittent carriage (OR = 0.85; CI: 0.75–0.97) compared to noncarriers, and current smokers experienced over two times the odds of intermittent carriage compared to noncarriers

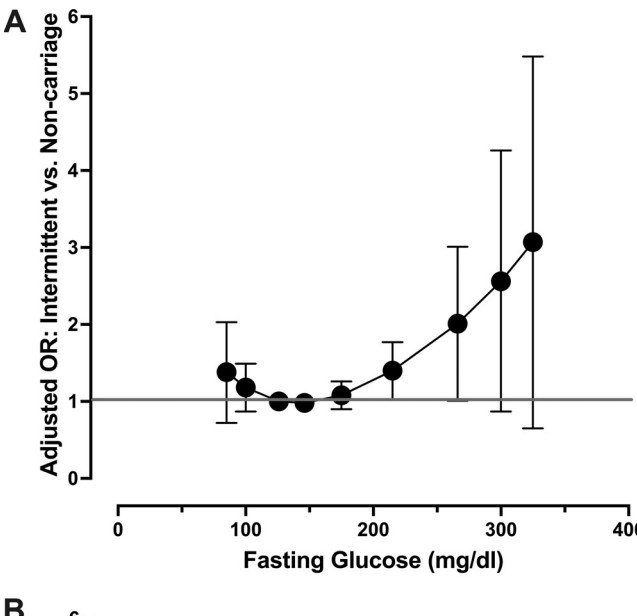

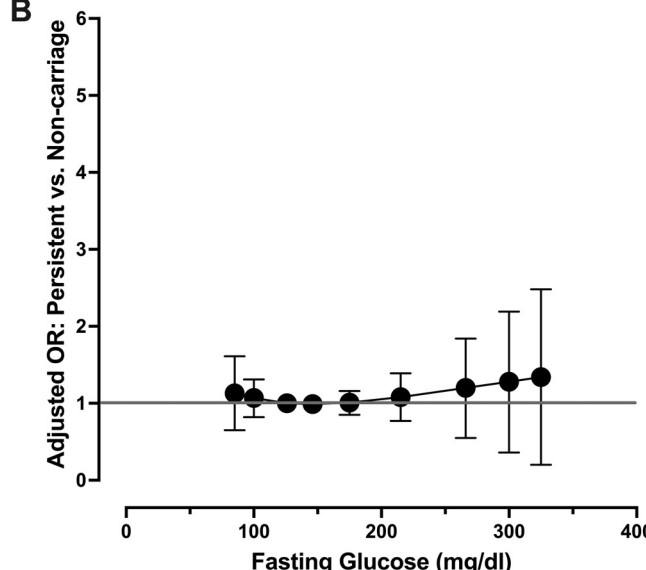

**FIG 1** Adjusted odds of intermittent (A) and persistent carriage (B) compared to noncarriers, respectively, across the spectrum of fasting glucose (mg/dL) in the primary cohort. The odds of intermittent carriage compared to noncarriage across the spectrum of fasting plasma glucose modeled with three-knot restricted cubic splines (knots at 125, 171, 225 mg/dL), with a value of 126 mg/dL (the clinical cut-point for diabetes) as the referent value, adjusted for use of DPP4 inhibitors, BMI, total cholesterol, and use of an antibiotic in the past 30 days (A). For comparison purposes, (B) models the same relationship for persistent carriers compared to noncarriers with fasting glucose forced into the model, adjusting for rounds of antibiotics used in the past year.

(OR = 2.05; CI: 1.30–3.25). These three variables were selected for inclusion in the multivariable intermittent carriage model and remained significant predictors of intermittent carriage with the magnitude of the adjusted effect slightly increasing for each predictor. While we did not have specific data regarding who in the replication cohort used DPP4 inhibitors, a medication that attenuated the effect of %HbA1c in the primary cohort, when use of any oral agent, the best approximation of use of DPP4 inhibitors available in this cohort, was forced into the multivariable model, the effect of % HbA1c on the odds of intermittent carriage was attenuated slightly (aOR = 1.24; CI: 1.09–1.41) but remained highly significant, while the effect of oral medication use remained non-significant (aOR = 1.44; CI: 0.80–2.59) (data not shown).

In the univariable model comparing persistent carriers to noncarriers, the number

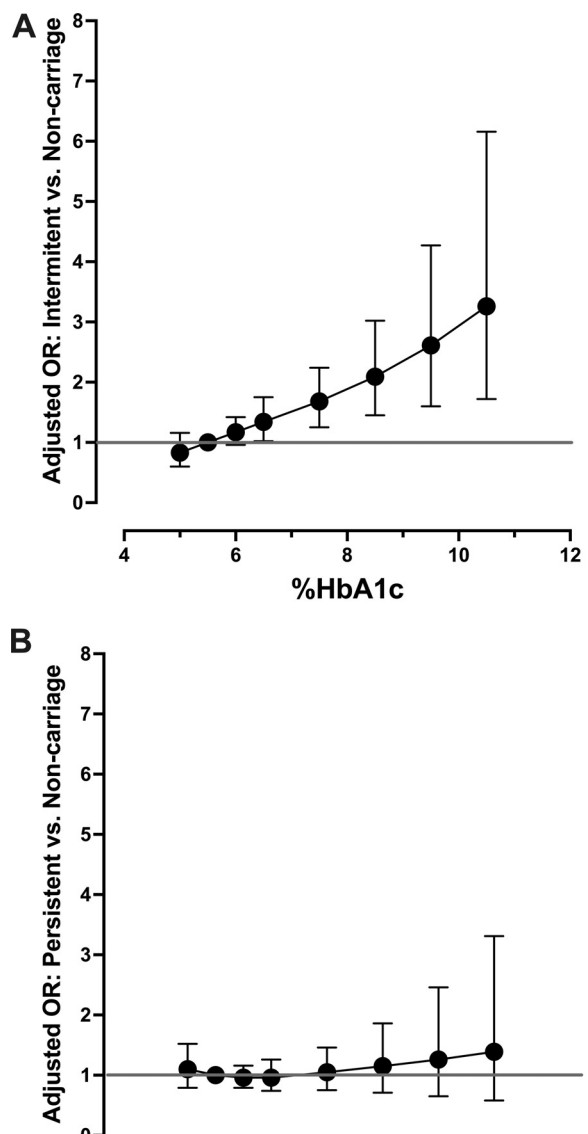

**FIG 2** Adjusted odds of intermittent (A) and persistent carriage (B) compared to noncarriers, respectively, across the spectrum of percent glycated hemoglobin A1c (%HbA1c) in the replication cohort. The odds of intermittent carriage compared to noncarriage across the spectrum of %HbA1c modeled with three-knot restricted cubic splines (knots at 5%, 5.5%, 7.3%), with 5.5% as the referent value and adjusted for use of age and smoking status (e.g., never, former, current) (A). For comparison purposes, (B) models the same relationship for persistent carriers compared to noncarriers with %HbA1c forced into the model and adjusted for rounds of antibiotics used in the past year, total serum cholesterol, and gender.

of rounds of antibiotics used in the past year was significantly associated with persistent carriage as it was in the primary study cohort. In addition, gender and total cholesterol were also associated with persistent carriage, with men having 2.37 times the odds (OR = 2.37; CI: 1.57–3.57) of persistent carriage compared to women, and a 10-unit increase in cholesterol associated with a 4% increase in the odds (OR = 1.04; CI: 1.01–1.08) of persistent carriage compared to noncarriers. In multivariable modeling, use of three or more rounds of antibiotics in the past year (aOR = 0.26; CI: 0.10–0.66) and gender (aOR = 2.23; CI: 1.46–3.41) both remained statistically significantly

**TABLE 3** Odds of intermittent and persistent carriage, respectively, compared to noncarriage in the replication cohort: univariable and multivariable models

| Characteristic | Intermittent carriage | | Persistent carriage | |
|---|---|---|---|---|
| | Univariable models | Multivariable model | Univariable models | Multivariable model |
| | Odds ratio (95% CI) | | | |
| Age[d] | 0.85 (0.75–0.97)[a] | 0.81 (0.71–0.93)[b] | 0.97 (0.85–1.11) | |
| Gender | | | | |
| Female | REF | | REF | |
| Male | 1.006 (0.67–1.52) | | 2.37 (1.57–3.57)[c] | 2.23 (1.46–3.41)[c] |
| Hemoglobin A1c (%) | 1.21 (1.07–1.36)[c] | 1.28 (1.13–1.44)[c] | 1.08 (0.94–1.25) | |
| Normoglycemia (<5.7%) | REF | | REF | |
| Prediabetes (5.7 ≤ % <6.5) | 0.85 (0.52–1.41) | | 0.94 (0.55–1.60) | |
| Diabetes (≥6.5%) | 1.62 (1.02–2.59)[a] | | 1.07 (0.61–1.90) | |
| Use of any diabetes medication | | | | |
| No | REF | | REF | |
| Yes | 1.49 (0.95–2.34) | | 0.73 (0.40–1.34) | |
| Use of insulin alone or in combination with an oral diabetes agent | | | | |
| No | REF | | | |
| Yes | 1.15 (0.48–2.75) | | Insufficient data | |
| Use of oral diabetes agent alone or in combination with insulin | | | | |
| No | REF | | REF | |
| Yes | 1.43 (0.88–3.32) | | 0.90 (0.47–1.74) | |
| Total cholesterol (mg/dL)[d] | 1.02 (0.99–1.06) | | 1.04 (1.01–1.08)[a] | 1.03 (0.99–1.07) |
| Antibiotic use in the past 30 days | | | | |
| No use | REF | | REF | |
| Use | 0.99 (0.61–1.61) | | 0.67 (0.37–1.23) | |
| Rounds of antibiotics used in the past 12 mo | | | | |
| 0 rounds | REF | | REF | REF |
| 1 round | 0.81 (0.49–1.31) | | 0.88 (0.53–1.45) | 0.87 (0.52–1.44) |
| 2 rounds | 0.77 (0.44–1.34) | | 0.56 (0.29–1.08) | 0.58 (0.30–1.14) |
| 3 or more rounds | 0.72 (0.40–1.29) | | 0.24 (0.09–0.62)[b] | 0.26 (0.10–0.66)[b] |
| Smoking status | | | | |
| Never smoker | REF | REF | REF | |
| Former smoker | 1.10 (0.68–1.80) | 1.22 (0.74–2.01)[b] | 1.59 (0.97–2.61) | |
| Current smoker | 2.05 (1.30–3.25)[b] | 2.07 (1.30–3.31) | 1.64 (0.96–2.84) | |
| Birthplace | | | | |
| USA | REF | | REF | |
| Mexico/Other[e] | 0.82 (0.55–1.23) | | 1.03 (0.65–1.63) | |
| Skin infection in the past 6 mo | | | | |
| No | REF | | REF | |
| Yes | 1.12 (0.45–2.82) | | 1.09 (0.37–3.00) | |
| Hospitalized in the past 12 mo | | | | |
| No | REF | | REF | |
| Yes | 1.42 (0.74–2.74) | | 1.06 (0.48–2.35) | |
| Household member hospitalized in the past 12 mo | | | | |
| No | REF | | REF | |
| Yes | 1.20 (0.66–2.16) | | 0.72 (0.34–1.54) | |
| No. in the household | 1.04 (0.94–1.15) | | 0.98 (0.88–1.10) | |
| No. of children in the home | 1.003 (0.89–1.14) | | 0.98 (0.86–1.12) | |

[a]*P* value < 0.05.
[b]*P* < 0.01.
[c]*P* < 0.001.
[d]OR interpreted as a 10–unit increase.
[e]1 born in El Salvador and 2 born in Honduras.

associated with persistent carriage, while the effect of total cholesterol was slightly attenuated and lost significance (aOR = 1.03; CI: 0.99–1.07).

The presence of interactions between all variables were assessed in both multivariable models, but none were present. Sensitivity analyses, including the 15 laboratory-presumptive persistent carriers removed from replication analyses due to incomplete *spa*-typing did not materially change the magnitude or statistical significance of any model results (data not presented).

As in the primary cohort, although the best model fit occurred when all continuous variables in the multivariable models were modeled linearly (compared to modeling with splines or a categorized version of the predictor), we chose to model, and plot the adjusted and unadjusted %HbA1c as splines to better appreciate the relationship between this variable and carriage phenotypes. Figure 2A depicts the odds of intermittent carriage compared to noncarriage across the spectrum of %HbA1c with a value of 5.5% as the referent value, adjusting for age and smoking status. Modeling of the splines shows a strongly linear significant increase in the odds of intermittent carriage with increasing %HbA1c. For comparison purposes, Fig. 2B models the same relationship for persistent carriers compared to noncarriers with %HbA1c forced into the model containing gender, total cholesterol, and rounds of antibiotics used in the past year. Fig. S3 shows the respective unadjusted plots of the odds of intermittent (Fig. S3A) and persistent carriage (Fig. S3B), each compared to noncarriage, respectively, with increasing %HbA1c.

## DISCUSSION

While previous studies have intimated at a link between *S. aureus* nasal carriage and diabetes status (5, 46–49), many of these studies were done in hospitalized or long-term care residents and without differentiation between intermittent and persistent carriers. The present study is the first to examine the odds of intermittent and persistent carriage across the glycemic spectrum utilizing multiple measures of glycemic status in two non-overlapping populations of non-hospitalized adults from the same community.

Host, bacterial, environmental, and genetic factors each play a role in defining the *S. aureus* nasal persistent, intermittent and noncarrier phenotypes (30, 44, 50–62); however, because persistent carriers differ in many ways from intermittent and noncarriers, it has previously been suggested that carriage phenotypes be dichotomized into persistent carriers versus all others (29). Specifically, analyses of antibody responses to *S. aureus* virulence factors suggests that the humoral responses of persistent carriers is distinct from those observed in non or intermittent carriers (29). There also appears to be an intimate association between the *S. aureus* carriage strain and their human hosts; for example, persistent carriage strains delay innate-immune activation compared to noncarrier strains and persistent carriage has been linked to altered innate immune responses, including decreased levels of $\beta$-defensin 3 (an antimicrobial peptide) and recognition of VITA-PAMPs (pattern-associated molecular patterns [PAMPs] that serve as microbial signatures of viability that activate the innate immune system) recognition (44, 50, 52). VITA-PAMPs are a PAMP subset that signals microbial life to the innate immune system such as bacterial mRNA, pyrophosphates, quorum sensing molecules, bacterial second messengers, and isopyrenoids resulting in heightened innate immune activation following recognition by cognate PRRs.

In addition, persistent carriers are more likely to be colonized with a single strain of *S. aureus* at high concentrations (29), and if decolonized from the nose, persistent carriers are more likely to be recolonized by their original colonizing strain (29, 63). Persistent carriers are also at a higher risk of infection, albeit with their own carriage isolate; however, disease severity in such carriers is diminished compared to disease presentation in non and intermittent carriers (64). For these reasons, intermittent carriers are often excluded from studies designed to undercover factors affecting *S. aureus* nasal carriage (52, 58, 65, 66).

Understanding risk factors for both persistent and intermittent carriage have implications not only for health and disease but also vaccine development particularly among those with diabetes (67). Specifically, diabetic foot ulcers are more frequently infected with *S. aureus* than with any other pathogen (22, 23) and understanding factors that may increase infection risk may inform efforts designed to mitigate the risk of *S. aureus* infections in the population.

In contrast to prior studies designed to identify factors affecting *S. aureus* carriage, data presented in this report demonstrates the impact of dysglycemia on intermittent but not persistent carriage. More specifically, measures of poorly controlled glycemia, i.e., increasing %HbA1c and fasting plasma glucose, significantly increases the odds of intermittent carriage without significantly affecting that of persistent carriage. We hypothesize that environmental changes in the host like poorly controlled blood sugar levels do not have discernible effects on the unique interactions that exist between persistent carriers and their respective strains. In contrast, increased blood glucose and possibly, the concomitant increase surface airway glucose concentration in nasal passages (68–70), can result in accompanying changes to host immune responses and the composition of the nasal microbiome that may explain the observed increased odds of intermittent carriage in this population. Supporting this hypothesis were parallel observations between other measures of metabolic disease and factors associated with increasing inflammation that correlated with intermittent but not persistent carriage, i.e., increasing BMI and cholesterol in primary cohort and increasing age and current smoking habits in the replication group. Additional support to the hypothesis that environmental changes affect intermittent, but not persistent carriage is the observation that DPP4 inhibitors, a diabetes therapy with known immunomodulatory properties, but not any other medication for the treatment of diabetes not only altered the significant association between increasing %HbA1c and risk for intermittent carriage in the primary cohort but also is itself a risk factor for intermittent carriage (71–73). Although metformin has been shown to increase risk of infection with *S. aureus*, its effects in the context of persistent and intermittent carriage other than the present report have not been explored (74, 75). It should be noted that the Garett et al. study was conducted *in vitro* (metformin added to cell culture media) or in mice (metformin delivered intraperitoneally) (74). This is a significant distinction since metformin not delivered orally does not have the same glucose-lowering effects in humans, suggesting the importance of drug delivery in the context of *S. aureus* carriage or infection making it difficult to extrapolate metformin effects in murine and cell culture models (76).

Although both the primary and replication cohorts were from the same source population, it is important to note the primary cohort was designed to include of 50% individuals with diabetes. In contrast, the replication cohort was a household convenience sample that resulted in the enrollment of younger participants with a lower prevalence of diabetes (18.03% [141/782] with either blood values diagnostic for diabetes or reporting a previous diagnosis of diabetes from a physician) and a lower proportion of diabetes medication use; however, our finding of increased odds of intermittent carriage with increasing blood glucose values was consistent across both populations.

Study strengths include the ability to examine two distinct cohorts with differing distributions of diabetes prevalence as well as substantial clinical and infection risk histories from the same source population in Starr County, Texas for associations between *S. aureus* nasal carriage and measures of glycemia. In addition, the availability of *spa* typing data on respective isolates allowed us to better define persistent carriage by the presence of *S. aureus* strains of the same *spa* type at two time points. This stricter definition of persistent carriage allowed us to rule out intermittent carriers mislabeled as persistent carriers colonized with distinct *S. aureus* isolates across time. Furthermore, since swabs were first plated on mannitol salt plates, the swab further enriched for 48 h, and replated at each time point, a persistent carrier could have 2–4 *spa*-typed colonies used to establish their carriage phenotype making it unlikely that these

participants' carriage phenotype was misclassified. In addition, with 150 different *spa* types (35 shared between the primary and replication cohorts; 58 and 57 unique *spa* types in the primary and replication cohorts, respectively) it would be unlikely that the 2–4 isolates collected (across two time points for persistent carriers) would have been the same by chance. Weaknesses include not having measurements of BMI or detailed medication data for the replication cohort, particularly with respect to diabetes medications like DPP4 inhibitors. Use of this drug attenuated the effect of %HbA1c in the primary cohort and was a significant predictor or intermittent carriage in that population; however, it is likely its use would be less prevalent in the replication cohort since the use of any diabetes medication was considerably less prevalent in this study group and because the replication cohort was younger and less affected by diabetes and DPP4 inhibitors are often a second-line therapy after other diabetes treatment failures (77, 78).

These results suggest that intermittent nasal carriage of *S. aureus* is a distinct phenotype from that of both persistent and noncarriage, and that such carriers possess their own unique constellation of risk factors contributing to this carriage phenotype. While the relationship between persistent carriers and their colonizing *S. aureus* strains appears to be less affected by the environment, persons with diabetes or clinical profiles associated with inflammation were more likely to intermittently carry *S. aureus* in their nares and may be at increased risk for infection with this organism.

## MATERIALS AND METHODS

**Study populations and design.** Starr County, Texas, one of 14 Texas counties on the U.S.-Mexico border, has a median 2019 household income of $29,294 and is among the most impoverished counties in the United States. with over one-third of the 64,633 residents, 96% of whom self-identify as Hispanic, living below the federal poverty level (79). The prevalence of chronic disease, specifically type 2 diabetes and obesity, exceeds U.S. averages (42) and the county ranks in the bottom 20% for primary health care accessibility in Texas, with only 16 primary health care providers in 2019 (ranked 191 of 215 Texas counties surveyed) (80). In the past 2 decades, Starr County has experienced a 74% increase in the prevalence of type 2 diabetes (42).

The data and the consenting procedures used for both the primary analysis and the replication cohort were approved by the University of Texas Health Science Center Institutional Review Board (HSC-SPH-06-0225) and written informed consent was obtained from all participants before enrollment. Participants in the primary analyses were identified and enrolled in a larger parent study occurring in Starr County, Texas from December 2010 through January 2014 as described previously (42). Briefly, participating adults aged 21 years or older, approximately 50% of whom were prevalent or incident cases of type 2 diabetes and 50% with no history or evidence of diabetes, were selected from a pool participating in a larger genome-wide association study (GWAS) (42). Subjects were recruited for a brief interview with 15 close-ended questions regarding age, sex, household size, number of rooms in the household, country of origin, anthropometric data, smoking status, history of antibiotic use, previous hospitalizations, and history of skin infections.

A similar questionnaire was given to adult participants in the GWAS replication cohort, a subject pool that was also used to study household *S. aureus* carriage profiles (45). Data were collected by consenting residents aged 18 years or older from sampled households in Starr County, Texas, between February 2013 and October 2014. Any participants with enrollment in both studies were subsequently removed from the replication cohort analyses.

In both studies, measures of plasma glucose were determined using a YSI 2300 STAT Plus Glucose & Lactate Analyzer (YSI Life Sciences, Yellow Springs, OH) in duplicate. Participants with a fasting plasma glucose greater than 160 mg/dL were not subjected to glucose challenge; consequently, measures of 2-h post-load glucose were not available for many participants and this measure was not considered in any analyses in the present study. %HbA1c and total serum cholesterol was measured using a Siemens DCA Vantage Analyzer point of care device (Malvern, PA). Standard clinical cut points were used for categorization of measures of glycemia (81). Body weight was measured on a Tanita Total Body Composition Analyzer (TBF-400, Arlington Heights, Illinois) with individuals in street clothing and no shoes. Height was obtained using a wall-mounted stadiometer. Body mass index (BMI) was calculated by dividing weight in kilograms by the square of height in meters. Antibiotic use data was collected by asking participants to recall the number of antibiotic courses taken in the past 12 months as well as the month and year of their most recent course of antibiotics. To determine the number of days since use, all participants were assumed to have taken their antibiotics on the first of the month. In the primary cohort, use of diabetes medication was collected by asking participants which medications, if any, they used, with response options, including: no use of any diabetes medication or use of metformin, sulfonylureas, thiazolidinediones, dipeptidyl peptidase 4 (DPP4) inhibitors, or insulin, respectively. In the replication study participants had the option to respond to a similar question with response options: no use of any diabetes mediation, use of insulin, oral medications, or both insulin and oral medications.

**Nasal swab sample collection.** For both the primary and replication studies, nasal swab samples were collected immediately superior the anterior nares using a dry, unmoistened sterile BBL CultureSwabs Liquid Stuart Swabs (BD Biosciences, Franklin Lakes, NJ). Swabs were inserted into each participant's nostrils approximately 1 in. from the edge of the anterior nares placing the swab in proximity with the inferior and middle concha and rolled several times. In both the primary and replication cohorts, a second nasal swab was obtained 11 to 17 days later and a follow-up interview collected as described (44, 82–84).

Bar-coded specimen tubes of nasal swab samples were stored at 4°C for no more than 3 days and shipped at 4°C to the University of Texas Health Science Center at the Houston School of Public Health. Upon arrival, swabs were streaked onto Mannitol Salt Agar (MSA) (Remel Inc., Lenexa, KS), a selective and differential media for isolating *S. aureus*, and incubated at 37°C for 48 h (85). Following primary plating, each swab was placed in tryptic soy broth (Remel Inc., Lenexa, KS), vortexed for 10 s to ensure any bacteria on the swab were released into the sterile media, incubated at 37°C for 48 h, and subsequently replated onto MSA. Presumptive *S. aureus* colonies growing on primary or secondary MSA plates were incubated at 37°C for 24 h on blood agar (tryptic soy agar (Remel Inc., Lenexa, KS) containing 5% defibrinated sheep's blood (Quad Five, Ryegate, MT)) to assess hemolysis and coagulase activity (BactiStaph Latex 450, Remel Inc., Lenexa, KS). Gram-positive and coagulase-positive colonies were classified as *S. aureus* and stored at −80°C in freezing media (15% glycerol, 85% tryptic soy broth) to await DNA extraction.

**DNA extraction and spa typing.** Frozen stocks of *S. aureus* nasal isolates were replated onto mannitol salt agar plates and incubated for 24h at 37°C. A single colony was introduced in 10.0 mL of tryptic soy broth and allowed to grow overnight at 37°C. DNA was extracted from 1.0 mL of enriched broth using the DNeasy blood and tissue kit (Qiagen, Valencia, CA) per the manufacturer's instructions. Extracted DNA was stored at −80°C until samples were shipped on dry ice to the University of Mississippi Medical Center to undergo *spa* genotyping. *Spa* genotyping and clonal complex determination was performed for all samples as described previously (45).

**S. aureus nasal carriage determination.** *S. aureus* nasal carriage phenotypes (persistent, intermittent, or noncarriage) were established by analyzing two nasal swabs collected 11–17 days apart using a modified 'two-culture rule' as described by our group previously (44, 82–84). The 'two-culture rule' as developed by Nouwen et al. (33) is a validated means of distinguishing between intermittent and persistent *S. aureus* colonization and has been employed in numerous populations (12, 57, 82, 86–90). Using this rule, persistent carriers are defined as those with culturable *S. aureus* from nasal swabs at both time points, intermittent carriers are those with a positive *S. aureus* culture from only one swab, and noncarriers are those for whom *S. aureus* could not be cultured from either swab as described previously (33, 44, 45, 83–85). Additionally, those with *S. aureus* detectable at both time points were defined as persistent carriers only if they were colonized with the same strain at both time points.

**Statistical analyses.** $\chi^2$ and ANOVA were used to compare carriage phenotypes across baseline characteristics for categorical and continuous variables, respectively. Univariate logistic regression was used to assess the association between baseline characteristics and intermittent and persistent carriage status, respectively, with noncarriers as the reference group in the primary study population. To account for the correlation inherent in the household sampling design used in the replication study, general estimating equations (family: binomial; link: logit) with exchangeable correlation matrix were used for all replication cohort models. Multivariable models were built using purposeful variable selection: univariable associations with a Wald $P$-value $< 0.20$ were entered in the multivariable model and covariates with Wald $P$-value of $< 0.10$ in the multivariable model were retained in the final model. Covariates considered included age, gender, birth country, diabetes medication use, recent or frequent antibiotic use, body mass index (not available in the replication study), fasting insulin (not available in the replication study), total serum cholesterol, smoking status, personal or family history of hospitalizations, personal history of skin infections, number of total household members, and number of children living in the household.

Correlations between predictors were determined using Pearson's $\chi^2$ and Spearman's rho correlation coefficient for categorical and continuous predictors, respectively, and the linearity assumption was assessed using restricted cubic splines (via likelihood ratio tests comparing the model containing splines against the nested linear model) and Akaike and Bayesian Information Criterions (AIC and BIC, respectively) in the primary study models and using Quasilikelihood under the Independence model Criterion (QIC/QICu) in the replication population models. The presence of interactions between all predictors included in final multivariable models were assessed using Wald $P$-values. To better observe the trajectory of the effect of diabetes measures on persistent and intermittent carriage, 3-knot restricted cubic splines were used, interpreted, and plotted using the 'xblc' postestimation command (91) in Stata 16 (Statacorp, College Station, Tx). A two-sided $P$-value $<0.05$ was considered statistically significant in all models. All analyses were performed in Stata 16.

**Data availability.** Data available upon request.

## SUPPLEMENTAL MATERIAL

Supplemental material is available online only.
**SUPPLEMENTAL FILE 1**, PDF file, 1.1 MB.

## ACKNOWLEDGMENTS

We are profoundly grateful to the amazing staff at the Starr County Health Studies field office; these studies would never take place without your hard work. Special

thanks also go out to all the participants in this and all our studies. Thank you for putting your trust in us to conduct this work.

We declare that we have no conflicts of interest.

E.L.B. and C.L.H. are the principal investigators and oversaw the study design and data collection. H.T.E. also participated in data collection, sample processing and data analyses, including statistical analyses. All authors (H.T.E., D.A.A., G.J., S.M.D., W.B.P., D.A.R., C.L.H., and E.L.B.) participated in manuscript preparation. All authors have read and approved the final version of the manuscript. D.A.R. oversaw the *spa* typing of collected *S. aureus* isolates.

Funding was received from National Institutes of Health grants R01DK116378 to E.L.B. and C.L.H. and grant 5T42OH008421 from NIOSH/Centers for Disease Control and Prevention to W.B.P.

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
