## [Reviewer comments · Microbiology Spectrum]

Microbiology Spectrum

Worsening Glycemia Increases the Odds of Intermittent but not Persistent *Staphylococcus aureus* Nasal Carriage in Two Cohorts of Mexican American Adults

Heather Essigmann, Craig Hanis, Stacia DeSantis, William Perkison, David Aguilar, Goo Jun, D. Robinson, and Eric Brown

Corresponding Author(s): Eric Brown, University of Texas-School of Public Health

Review Timeline:

Submission Date:	January 6, 2022
Editorial Decision:	March 29, 2022
Revision Received:	April 20, 2022
Accepted:	April 21, 2022

Editor: Christopher LaRock

Reviewer(s): Disclosure of reviewer identity is with reference to reviewer comments included in decision letter(s). The following individual involved in review of your submission has agreed to reveal their identity: Anne-Sofie Furberg (Reviewer #2).

Transaction Report:

DOI: <https://doi.org/10.1128/spectrum.00009-22>

March 29, 2022

Dr. Eric L Brown
University of Texas-School of Public Health
Center of Infectious Diseases
1200 Herman Pressler Drive
Houston, TX 77030

Re: Spectrum00009-22 (Worsening Glycemia Increases the Odds of Intermittent but not Persistent Staphylococcus aureus Nasal Carriage in Two Cohorts of Mexican American Adults)

Dear Dr. Eric L Brown:

My sincerest apologies for the delay, we had issues with reviewer completion but secured a quick backup. I am happy the news is quite positive: On the basis of recommendations from expert reviewers in the field, I have determined that your manuscript requires only minor edits before acceptance. Both reviewers found the study to overall be of high interest to the field, but had a few questions for clarification. This can be addressed in a text-only revision. In your revision, be sure to conform to journal guidelines for formatting requirements.

Thank you for submitting your manuscript to Microbiology Spectrum. As you will see your paper is very close to acceptance. Please modify the manuscript along the lines I have recommended. As these revisions are quite minor, I expect that you should be able to turn in the revised paper in less than 30 days, if not sooner. If your manuscript was reviewed, you will find the reviewers' comments below.

When submitting the revised version of your paper, please provide (1) point-by-point responses to the issues I raised in your cover letter, and (2) a PDF file that indicates the changes from the original submission (by highlighting or underlining the changes) as file type "Marked Up Manuscript - For Review Only". Please use this link to submit your revised manuscript. Detailed instructions on submitting your revised paper are below.

Link Not Available

Sincerely,

Christopher LaRock

Reviewer comments:

Reviewer #1 (Comments for the Author):

All the methods have been applied rigorously and with adequate replicability. The data have been supported by standard statistical analysis.

Reviewer #3 (Comments for the Author):

In "Worsening Glycemia Increases the Odds of Intermittent but not Persistent Staphylococcus aureus Nasal Carriage in Two Cohorts of Mexican American Adults", Essigmann et al nasal carriage of *S. aureus* in an adult Mexican American population. This population may have an increased disease burden, but is underrepresented in studies. This study is well-conducted and controlled and will be of use to the field. I have but a few minor comments
When discussing association of drugs with *S. aureus*, a broader audience would benefit with reference to example where a

molecular basis for impact is known or hypothesized (doi 10.1002/dmrr.2975 and 10.1016/j.chom.2010.10.005, but there may be better)

Line 286: Type "s" to "is"

Line 291: greater description of VITA-PAMP (perhaps including examples) needed

Lines 338-340: Since spa typing was only performed on a single colony from each swab, can the authors exclude the possibility that undersampling could exclude some persistent carriers as intermittent carriers? What is the diversity of spa types found in this population? Low diversity would be expected to give a high co-incidence making intermittent carriage appear persistent. If these are possible, they should be discussed more.

Preparing Revision Guidelines

- point-by-point responses to the issues I raised in your cover letter
- Upload a compare copy of the manuscript (without figures) as a "Marked-Up Manuscript" file.
- Each figure must be uploaded as a separate file, and any multipanel figures must be assembled into one file.
- Manuscript: A .DOC version of the revised manuscript
- Figures: Editable, high-resolution, individual figure files are required at revision, TIFF or EPS files are preferred

Please return the manuscript within 60 days; if you cannot complete the modification within this time period, please contact me. If you do not wish to modify the manuscript and prefer to submit it to another journal, please notify me of your decision immediately so that the manuscript may be formally withdrawn from consideration by Microbiology Spectrum.

The authors have taken up a research on Staph carriage among Diabetic patients. They have very well established that patients with poorly controlled diabetes are at a higher risk of acquiring Staph infections. Their experimental design is apt and the results have been supported with statistical evidences. I recommend the acceptance of this manuscript in its present form.

Response to Reviewers

Reviewer #3 (Comments for the Author):

In "Worsening Glycemia Increases the Odds of Intermittent but not Persistent Staphylococcus aureus Nasal Carriage in Two Cohorts of Mexican American Adults", Essigmann et al nasal carriage of *S. aureus* in an adult Mexican American population. This population may have an increased disease burden, but is underrepresented in studies. This study is well-conducted and controlled and will be of use to the field.

I have but a few minor comments.

When discussing association of drugs with *S. aureus*, a broader audience would benefit with reference to example where a molecular basis for impact is known or hypothesized (doi 10.1002/dmrr.2975 and 10.1016/j.chom.2010.10.005, but there may be better).

Response:

We thank the reviewer for the constructive critique of our manuscript. In reference to the point regarding drug associations with *S. aureus*, we have added the section below (**lines 325-332**) and added the referenced citation (doi 10.1002/dmrr.2975) (new reference #76). Various studies have identified genetic and environmental risk factors for *S. aureus* carriage (persistent and intermittent). We did not include the reviewer's second manuscript regarding the use of statins in the context of *S. aureus* since the effects of statins alter neutrophil function; an immune component not involved in nasal carriage.

New Text: Although metformin has been shown to increase risk of infection with *S. aureus*, it's effects in the context of persistent and intermittent carriage other than the present report have not been explored (74, 75). It should be noted that the Garrett *et al.* study was conducted *in vitro* (metformin added to cell culture media) or in mice (metformin delivered intraperitoneally) (74). This is a significant distinction since metformin not delivered orally does not have the same glucose-lowering effects in humans, suggesting the importance of drug delivery in the context of *S. aureus* carriage or infection making it difficult to extrapolate metformin effects in murine and cell culture models (76).

Line 286: Type "s" to "is"

Response:

Corrected.

Line 291: greater description of VITA-PAMP (perhaps including examples) needed.

Response:

VITA-PAMPs have been described in greater detail.

Lines 290-293:

VITA-PAMPs are a PAMP subset that signals microbial life to the innate immune system such as bacterial mRNA, pyrophosphates, quorum sensing molecules, bacterial second messengers, and isopyrenoids resulting in heightened innate immune activation following recognition by cognate PRRs.

Lines 338-340: Since spa typing was only performed on a single colony from each swab, can the authors exclude the possibility that undersampling could exclude some persistent carriers as intermittent carriers? What is the diversity of spa types found in this population? Low diversity would be expected to give a high co-incidence making intermittent carriage appear persistent. If these are possible, they should be discussed more.

Response:

This point has been addressed in both the Discussion. Although stated in the materials section, it was not clear and therefore mentioned again in the Discussion. When a sample is tested for the presence of *S. aureus*, the swab is used to streak a mannitol salt plate and then incubated in media for 48 h to enrich for any *S. aureus* missed during the first plating. This enriched media is plated on a new mannitol salt plate. What this means is that a persistent carrier could have two *S. aureus* isolates spa-typed at each timepoint for a total of 4 (or a minimum of 2). We also included the number total spa types (n=150) across both populations.

The new text is below:

Lines 348-354 (Discussion):

Furthermore, since swabs were first plated on mannitol salt plates, the swab further enriched for 48 h, and replated at each time point, a persistent carrier could have 2-4 spa-typed colonies used to establish their carriage phenotype making it unlikely that these participants' carriage phenotype was misclassified. In addition, with 150 different spa types (35 shared between the primary and replication cohorts; 58 and 57 unique spa types in the primary and replication cohorts, respectively) it would be unlikely that the 2-4 isolates collected (across two time points for persistent carriers) would have been the same by chance.

April 21, 2022

Dr. Eric L Brown
University of Texas-School of Public Health
Center of Infectious Diseases
1200 Herman Pressler Drive
Houston, TX 77030

Re: Spectrum00009-22R1 (Worsening Glycemia Increases the Odds of Intermittent but not Persistent Staphylococcus aureus Nasal Carriage in Two Cohorts of Mexican American Adults)

Dear Dr. Eric L Brown:

Your manuscript has been accepted, and I am forwarding it to the ASM Journals Department for publication. You will be notified when your proofs are ready to be viewed.

Sincerely,

Christopher LaRock
Editor, Microbiology Spectrum
